# Mesenchymal Stem Cell-Based Therapy as an Alternative to the Treatment of Acute Respiratory Distress Syndrome: Current Evidence and Future Perspectives

**DOI:** 10.3390/ijms22157850

**Published:** 2021-07-22

**Authors:** Silvia Fernández-Francos, Noemi Eiro, Natalia González-Galiano, Francisco J. Vizoso

**Affiliations:** 1Research Unit, Fundación Hospital de Jove, Av. Eduardo Castro, 161, 33290 Gijón, Spain; silviafernandezfrancos@gmail.com (S.F.-F.); noemi.eiro@hospitaldejove.com (N.E.); 2Department of Anesthesiology, Fundación Hospital de Jove, Av. Eduardo Castro, 161, 33290 Gijón, Spain; 3Department of Internal Medicine, Fundación Hospital de Jove, Av. Eduardo Castro, 161, 33290 Gijón, Spain; natalia.gg.88@gmail.com; 4Department of Surgery, Fundación Hospital de Jove, Av. Eduardo Castro, 161, 33290 Gijón, Spain

**Keywords:** COVID-19, pneumonia, acute respiratory distress syndrome, acute lung injury, mesenchymal stem cells, exosomes, conditioned medium

## Abstract

Acute respiratory distress syndrome (ARDS) represents a current challenge for medicine due to its incidence, morbidity and mortality and, also, the absence of an optimal treatment. The COVID-19 outbreak only increased the urgent demand for an affordable, safe and effective treatment for this process. Early clinical trials suggest the therapeutic usefulness of mesenchymal stem cells (MSCs) in acute lung injury (ALI) and ARDS. MSC-based therapies show antimicrobial, anti-inflammatory, regenerative, angiogenic, antifibrotic, anti-oxidative stress and anti-apoptotic actions, which can thwart the physiopathological mechanisms engaged in ARDS. In addition, MSC secretome and their derived products, especially exosomes, may reproduce the therapeutic effects of MSC in lung injury. This last strategy of treatment could avoid several safety issues potentially associated with the transplantation of living and proliferative cell populations and may be formulated in different forms. However, the following diverse limitations must be addressed: (i) selection of the optimal MSC, bearing in mind both the heterogeneity among donors and across different histological origins, (ii) massive obtention of these biological products through genetic manipulations of the most appropriate MSC, (iii) bioreactors that allow their growth in 3D, (iv) ideal culture conditions and (v) adequate functional testing of these obtaining biological products before their clinical application.

## 1. Introduction

The lung is a complex organ directly and chronically exposed to outdoor air pollution, facing microorganisms and noxious agents. It is composed of a large and strongly vascularized epithelial surface, whose main functional challenge is efficient gas exchange.

Lung diseases are a steadily growing global health concern and a major cause of death and morbidity worldwide. Thus, 3.2 million people were reported to have died of chronic obstructive pulmonary disease (COPD) in 2017 [1] (5.7% of global mortality), and it is projected to be the third-leading cause of death worldwide by 2030. In this context, acute lung injury (ALI) represents, nowadays, a global community medical concern caused by several insults such as viral or bacterial infections inter alia. Acute respiratory distress syndrome (ARDS) is one frequent and severe form of ALI according to the Berlin definition 2012 [2], which, despite improvements in supportive care and antibiotic use, is associated with a high mortality (30–40%). The pathogenesis of ARDS is conditioned by the imbalance of the immune response, the inflammation-induced disruption of the alveolar endothelial/epithelial barrier, the activation of clotting and proliferation of fibroblasts, particularly in the interstitium, and fibrosis.

Today, we are witnessing a dramatic example of this critical situation due to the pandemic by coronavirus disease-2019 (COVID-19), which highlights that innovative therapeutic strategies are urgently necessary. There are several data supporting a potential therapeutic role of mesenchymal stem cells (MSC) in lung diseases. Interestingly, it has been reported that a phenotype alteration of pulmonary MSC associated to lung pathology, such as COPD and bronchopulmonary dysplasia, has effects related to aging and ALI [3,4]. Experimental lung disease in vivo models such as COPD, asthma, bronchopulmonary dysplasia, idiopathic pulmonary fibrosis and acute lung injury show the therapeutic efficacy of MSC [5] or their exosomes [6]. Furthermore, clinical phase I and phase II studies based on MSC administration have shown preliminary safety outcomes in patients undergoing these processes [7,8]. Given the tremendously serious and imperative nature of the situation of the coronavirus pandemic by severe acute respiratory syndrome coronavirus 2 (SARS-CoV-2), early and urgent clinical trials demonstrated the safety and efficacy of MSC in COVID-19 patients, being this latter attributed to their anti-inflammatory mechanism against the cytokine storm associated to COVID-19 [9], and new clinical trials are underway by using MSC from several origins and with different administration routes [10]. Based on these introductory results, the treatment of critically ill patients is considered under compassionate use protocols [11]. In this context, the “Italian College of Anesthesia, Analgesia, Resuscitation and Intensive Care” have reported guidelines to treat COVID-19 patients.

The aims of the present review consist of exposing the evidence on the potential therapeutic use of MSC for ARDS, their mechanisms of action, clinical trials, limitations of MSC-based therapies and future perspectives of their secretome or secretome-derived products.

## 2. Acute Respiratory Distress Syndrome

Five decades after its first description in 1967, ARDS remains a devastating condition among critically ill patients, representing an important public health problem globally. The use of protective ventilation strategies (e.g., low tidal volume ventilation and prone position) has mitigated the damaging effects of mechanical ventilation. However, all pharmacological or cell-based therapy interventions have failed thus far, and mortality remains as high as 40%.

There is no large international cohort study hitherto showing updated rates of ARDS incidence and mortality, although incidence of ARDS in the United States has been reported as 10 times higher than in Europe [12]. Depending on the diagnostic criteria and study design, the prevalence of ARDS in the United States is estimated to range from 5 to 35 cases/100,000 individuals annually. Additionally, among Intensive Care Units (ICU) in 50 countries, the period prevalence of ARDS was 10.4% of ICU admissions [13].

The wide range of triggers leading to ARDS development acts as a major confounder that may have hindered effective drug therapies up until now. These causes include direct pulmonary injuries (bacterial and viral pneumonia, aspiration of gastric contents, pulmonary contusion or inhalation injury) and indirect extrapulmonary lesions (sepsis, which stands out as the most frequent etiology, appearing in 79% of ARDS episodes, severe trauma, transfusions, pancreatitis, drug reactions, burns and cardiopulmonary bypass surgery, etc.).

Once turned on, the pathophysiology of ARDS results from a complex interaction between the immune system and the alveolar-capillary barrier, which culminates in fibroproliferation as the final stage of the tissue defense response [14]. This defense is organized around the following three pillars that are activated simultaneously: (i) acute inflammatory reaction: increased microvascular permeability, impairing the pulmonary surfactant secretion and alveolar fluid clearance, an intensified expression of adhesion molecules and extravasation of leukocytes and their release products, which cause tissue destruction; (ii) hemostasis: intravascular coagulation and extravascular fibrin deposition and the inhibition of fibrinolysis) and (iii) tissue restoration: the regeneration of endothelial and epithelial type I/II cells in the alveoli, fibroproliferation and an accumulation of the extracellular matrix. 

Lung injury and tissue repair processes often coexist, although not uniformly in all patients. Understanding the differences in the recovery course and why some patients develop more fibrosis than others will help identify new alternatives as treatment strategies. 

These pathological features correspond to some generalized clinical characteristics of ARDS (bilateral opacities on chest images, respiratory failure not fully explained by heart failure, and altered oxygenation), as reflected in the Berlin 2012 definition.

However, the predictive ability of applying this definition is limited due to physician subjective interpretation, different disease pathways and genetic and acquired factors, capable of affecting individual susceptibility and, consequently, generating an unpredictable trajectory of disease. Indeed, the clinical outcomes of ARDS are variable among patients, where some recover completely, some survive with a permanent decreased lung function, while others die [15].

In surviving patients, there is significant morbidity associated with cognitive impairment, residual pulmonary fibrosis, neuromuscular weakness and enduring neuropathy and myopathy. Even 5 years after ARDS, some of these conditions persist, raising the costs of healthcare resources. At present, there is no available pharmacotherapy arsenal and no valid standard of care approach in the treatment of ARDS, despite the advances in the understanding of its biology and physiopathology. The handling options are primarily limited to supportive measures; therefore, pioneering methodologies will be needed to move towards an effective therapy in ARDS [15].

## 3. Mesenchymal Stem Cells

MSC were firstly described by Friedenstein et al. [16] in 1970, as a “colony forming unit-fibroblast” present in stroma rodent’s bone marrow. Bone marrow MSC (BM-MSCs) are the most extensively studied MSC. After many decades of research, MSC were found in the stroma of other locations, such as adipose tissue (AT), umbilical cord (UC), Wharton’s jelly (WJ), uterus, placenta, amniotic fluid, lung, cartilage synovial fluid or membrane, tonsil, dermis, skeletal muscle, periosteum, periodontal ligament dental pulp or peripheral blood [17]. MSC have a fibroblastic spindle-shape morphology in standard culture media. The “International Society for Cellular Therapy” has established an universal criteria for an MSC definition in 2006 as follows: (i) MSC must display plastic-adherent capacities; (ii) a simultaneous expression of stromal markers (CD29, CD44, CD73, CD90 and CD105), but the absence of hematopoietic (CD45 and CD14) or endothelial (CD31 and CD34) markers and HLA-DR surface molecules and (iii) an in vitro differentiation potential into osteoblasts, adipocytes and chondroblasts [18].

MSC, although in minimal quantities, are present in many organs of the body, including the lungs, and participate and control tissue renewal. Indeed, lung gas exchange and host defense require the integrity of its epithelium and its dynamic interplay with neighboring mesenchyme. Lung cell refurbishment is usually slow compared to other adult organs such as the skin and gut. However, meaningful regeneration and repair are possible after physiological insults, including severe respiratory infection, thanks to the role of resident stem and progenitor cells [19]. MSC also have a function as active sentinels and regulators in tissue homeostasis. In fact, MSC have a regulatory role in basic biological processes, through their antimicrobial effects, potent anti-inflammatory properties and ability to control cell proliferation, apoptosis, angiogenesis or oxidative stress. These actions, for which MSC influence their tissue microenvironment, are possible through the secretion of soluble bioactive molecules and/or exosomes. 

In addition, it has been reported that in many degenerative or autoimmune diseases (such as rheumatoid arthritis [20], systemic lupus erythematosus [21], diabetes mellitus [22] or psoriasis [23]) and in processes associated with aging [24,25], there could be a depletion or poor function of these cells. Subsequently, the concept of restoring normal physiological function by using allogenic MSC has inspired many researchers around the world [26]. Another special property that makes MSC attractive for cell therapies is their homing effect. MSC homing refers to their ability to spontaneously migrate to the injured region when the body is damaged. MSC have been proven to have the competency to home to hurt areas after transplantation in in vivo studies. In vitro studies determined that the expression of chemotactic cues from injured tissues stimulate the MSC attraction. These recognized signals include TNF-α [27], PDGFα [28], IGF-1 [29], HGF and EGF [30], as well as the expression of the chemokines for which MSC have receptors (CXC1, 4, 7, 9 and 10, CXC4 and 6 and CXCL12) [31]. In addition, MSC secrete matrix metalloproteases (MMP), such as MMP1 (also named interstitial collagenase), which degrade the structural components that allow for the motility of MSC across the endothelial basement membrane.

On the other hand, additional studies to boost the tropism for the site of MSCs could offer the benefit of reducing the number of MSC required to reach therapeutic effects [32]. To that end, different methods have been proposed, such as genetic manipulation, the direct administration of MSC in the target tissue, alterations of cell surface, in vitro priming/preconditioning. At the same time, MSC, due to their inherent homing/targeting capacity, also offer the possibility of being loaded with certain drugs and taking advantage of their accumulation and penetration properties into tumors. Thus, this strategy could address the non-selective cytotoxicity of chemotherapeutic agents. In this sense, a “trojan horse” biomimetic transport stratagem using MSC for photodynamic and photothermal therapies against lung melanoma metastasis has been recently reported [33].

Over 1000 clinical studies have been performed up to date investigating the therapeutic potential of MSC. The safety profile and the efficacy of MSC therapy have been demonstrated in phase I, II and III clinical trials for numerous pathologies, including autoimmune and inflammatory diseases, allotransplant rejection, myocardial infarction, spinal cord injury, bone diseases, degenerative disorders, extensive burns and severe chronic wounds or grave pneumonia [34]. The functional improvement observed in these clinical studies after MSC infusions has been mainly attributed to MSC’s aptitude to interrelate with immune cells and secrete a variety of paracrine factors, which ultimately result in immunomodulation. In recognition of the beneficial effects of MSC to treat anal fistulas in Crohn’s disease and graft-vs.-host disease, the Food and Drugs Administration (FDA) and the Spanish Agency for Medicines and Health Products (AEMPS) have granted the commercial approval of some MSC-based products targeting these specific indications (Alofisel and Remestemcel-L, respectively).

## 4. Mechanisms for Which MSC Improve ALI or ARDS

Among the possible mechanisms for which MSC can improve both ALI or ARDS are antimicrobial, anti-inflammatory, regenerative, anti-oxidative stress, angiogenic and antifibrotic signals (Figure 1).

### 4.1. Antimicrobial

The expression of interferon-stimulated genes (ISG) (CCL2, IFI6, ISG15, PMAIP1, SAT1 and p21/CDKN1A) by MSC usually makes them resistant to viral infections, such as dengue virus, Ebola virus, influenza A virus and West Nile virus, inter alia. However, the infection capacity of SARS-CoV-2 may be due to IFN-independent pathways that could regulate the expression of currently known ACE2 or TMPRSS2 virus input receptors [35,36]. Continuing with the preceding, the very weak or non-existent expression of ACE2 and TMPRSS2 receptors on the surface of MSC, has been categorically demonstrated, making MSC theoretically resistant to SARS-Cov-2 infection. For that reason, alternative routes of entry for SARS-CoV-2 in MSC cannot be excluded; therefore, more studies are needed to testify the evidence that human MSC are not lenient for SARS-CoV-2 infection [37]. In addition, soluble proteins from MSC, such as IL-10, PGE2 and TNF-α, have a defensive effect against microorganisms [38]. Alternatively, MSC might act directly through the secretion of antimicrobial peptides such as cathelicidin, defensins, cystatin C, elafin and lipocalin 2. These are small, evolutionarily conserved effector molecules with a size ranging from 10 to 150 aminoacids, which mediate antimicrobial cell killing by inhibiting DNA or RNA synthesis and interacting with certain intracellular targets [39]. The main mechanism of action is reported to take place through a cleavage product of the cathelicidin, LL-37, which has a broad range of antibacterial activity against both Gram-negative and Gram-positive bacteria and antiviral actions. Interestingly, LL-37 is also functionally dependent on TLR modulation, which trigger the machinery of MSC to activate both the inflammatory NF-κB pathway and the interferon regulatory factor, which are fundamental to fight viral infections [40]. TLR4-primed MSC following microbial molecules recognition secrete chemokines such as MIP-1α and MIP-1β, RANTES, CXCL9, CXCL10 and CXCL11, as well as IL-6, IL-8, MIF and GM-CSF, which promote neutrophils and monocytes recruitment [41].

### 4.2. Anti-Inflammatory Effect 

It is known that under an inflammatory microenvironment MSC are activated to secrete high levels of soluble anti-inflammatory factors, such as COX2, IDO, NO, TGF-β1, PGE2 and HLA-G5 [42]. Analogously, some researchers have also pointed out that MSC have beneficial effects on experimental models for ALI, such as reducing pulmonary edema and maintaining alveolar-endothelial barrier homeostasis, actions mediated by TGF-β1, IDO, NO, IL-1RA, KGF and IL-10 [43]. Consequently, MSC have the following immunoregulatory effects on all immune cell types: a suppressive effect on T cells proliferation, inhibition of B cells, natural killer and dendritic cells function and polarization or reprogramming macrophages toward IL-10 producing anti-inflammatory phenotype. 

### 4.3. Regenerative Effect

Tissue regeneration includes sequential processes such as migration, anti-inflammatory/immunomodulatory actions, accelerated re-epithelization, improvement in extracellular matrix (ECM) production and remodeling. The factors responsible for these mechanisms encompass inflammatory proteins (IL-1, 6, 8-11, 13, PGE2, MCP-1), growth factors (EGF, KGF, TGF-β, HGF, FGF, VEGF, GF-1, PDGF, BNDF, NGF-3, IG-CSF GM-CSF and PGE2) and ECM proteins (MMP-1, 2, 3, 7, TIPM-1 y 2, ICAM, collagens, laminin, elastin and decorin) [17]. In addition, increased angiogenesis was proposed as another of the main mechanisms of regenerative effect from MSC by their paracrine action. MSC secrete molecular factors that improve the proliferation and migration of endothelial cells, such as VEGF, PDGF, ANG-1 y 2, EGF, FGF, TGF-β1, TGF-α, MCP-1, CXCL5 and MMP. Regarding the specific regeneration of pulmonary alveoli, it is known that MSC secrete plenty of molecules with paracrine effects that promote this process, including ANG-1, HGF, EGF, KGF and VEGF.

### 4.4. Antifibrotic Effect

In ALI pathological processes, fibrosis progresses and densifies. This scenario causes scar tissue formation as a result of the excessive deposition of ECM proteins, such as fibronectin, collagen I and collagen III. Consequently, lung tissue fails to develop its regenerative ability to assume its biological functions [44].

In vivo studies of different experimental models demonstrate the antifibrotic effects of MSC via the secretion of soluble factors such as MMP-1, HGF, TGF-β3, TNF-α, IL-10, VEGF and HGF, and also, fibrosis changes TGF-β1-induced suppression, making the TGF-β1/Smad pathway a principal pathogenic instrument in tissue fibrosis. In addition, BM-MSC-derived exosomes [45] and secretome [46] can reduce organ fibrosis. Recently, experimental evidence indicates that a transplantation of MSC could effectively treat pulmonary fibrosis as opposed to the administration of antifibrotic drugs such as nintedanib or pirfenidone, with a significantly better result in lung volume, pathological changes, lung function and blood oxygen saturation [47].

### 4.5. Other Effects 

Other effects for which MSC may improve ALI include the prevention of the epithelial-mesenchymal transition (EMT) of alveolar epithelial cells in the context of lung injury, as well as the anti-oxidative stress effect, anti-apoptotic actions and improvement of alveolar fluid clearance. EMT is a physiopathological process where epithelial cells modify their adhesion molecules expressed on the cell surface and develop mesenchymal phenotype and properties, allowing them to acquire a migratory and invasive behavior. This change is reflected in the reduction in surfactant production and the increase in metalloprotease secretion, deteriorating lung function. This phenotype conversion is executed in response to multiple extracellular ligands triggering signaling cascades that, directly or indirectly, downregulate the expression of adhesion molecules. 

Additionally, the cell environment and pleiotropic signals such as reactive oxygen species intervene in different signaling routes leading to EMT. Indeed, it has been indicated that exposure to hypoxia, a condition usually found in alveoli during ALI or chronic lung injury with lung remodeling, could foster phenotypic changes in alveolar epithelial cell lines compatible with EMT. 

Although stress exists in every cell, the pathological state depends on how much the disruption of the balance between free radicals and antioxidants takes place. Increased ROS (reactive oxygen species) levels in cells can cause damage by oxidizing DNA, proteins and lipids and by launching cell stress-activating pathways. Damaged cells disrupt ECM, promote inflammatory mediators’ release, nutrient deficiencies and hypoxic conditions due to poor blood supply, creating a deleterious microenvironment in the affected region. This pathological process conduces to inflammation, apoptosis, fibrosis and can aggravate the pathology driving to organ deficiency. Several studies have demonstrated that MSC show noteworthy low susceptibility to the harmful effect of reactive species, which is related to constitutively expressed antioxidant enzymes (SOD1, SOD2, catalase), glutathione peroxidase (Gpx) and high levels of the antioxidant glutathione (GSH). Moreover, MSC express heat-shock protein 70 (HSP70) and sirtuin (SIRT), which may also contribute to the resistance to oxidative stress [48]. The antioxidant effects of MSC therapy have been observed in vivo in many disease models, including lung injury. There are data revealing numerous mechanisms by which MSCs respond to oxidative stress, including the direct scavenging of free radicals, boosting endogenous antioxidant defenses, immunomodulation through reactive oxygen species suppression, altering mitochondrial bioenergetics and donating functional mitochondria to hurt cells [49].

Apoptosis is a dynamic process that strictly controls the rate of cell division and death and induces a suicide program with an enhanced DNA degradation, a swelling nucleus membrane, a shrinking cytoplasm and, ultimately, cell death. MSC were shown to be required for withholding apoptosis by upregulating the proliferation indicator Ki67 and anti-apoptotic markers BCL2 and SURVIVIN, and by downregulating the markers of apoptosis TUNEL, annexin V, CASPASE3 and CASPASE9 [50]. Apoptosis of both resident and immune cells contribute to ARDS progression. The in vivo administration of MSC to ARDS models can reduce the apoptotic cell counts in the lung tissues and distal organs. In fact, MSC have been reported to lessen apoptosis of the different cell types of lung tissue, such as alveolar epithelial and endothelial cells, by the secretion of KGF, HGF and ANG-1 [51], and alveolar macrophage apoptosis induced by lipopolysaccharide (LPS), which may be partially mediated by the inhibition of the Wnt/β-catenin pathway. In addition, it is a remarkable recent finding suggesting that MSC may reduce the apoptosis of lymphocytes induced by SARS-CoV-2 infection, which leads to lymphocytopenia in critically ill patients with COVID-19 pneumonia [52].

Excessive alveolar and interstitial fluid impairs surfactant concentration and gas exchange. Therefore, this pathological fluid removal facilitates lung function recovery in ARDS. There are data indicating that MSC can promote the alveolar fluid clearance by the secretion of paracrine factors, such as ANG-1 and FGF7 (or KGF). These growth factors mediate the maintenance of the apical membrane of epithelial cells, restoring the cell sodium channels, thereby normalizing the elimination of the alveolar fluid. In addition, experimental in vivo studies showed that the administration of MSC increases the alveolar fluid extraction in ARDS models or in ex vivo perfused human lung injured by endotoxin or Escherichia coli [53]. These effects have been related to the secretion of FGF-7 and extracellular vesicles (EV). 

## 5. Clinical Experience of MSCs as Therapy for ARDS

MSC clinical studies in ARDS are based on in vivo experimental research. In turn, clinical trials have been conducted to explore the therapeutic potential of MSC in ALI, as well as phase I and phase II clinical studies that demonstrate preliminary safety results in patients suffering from several lung diseases.

For its part, preclinical studies evaluated the treatment of ALI with MSC from different sources (B.M., A.T. and U.C.) [54] in different experimental lung inflammation models (LPS, influenza, Klebsiella, Pseudomonas aeruginosa or Escherichia coli), with diverse application routes (intravenous, intratracheal or intranasal) and a differing dose of cells. The results of these studies demonstrated that an MSC administration reduced the bacterial load, lung injury, inflammation (inflammatory cell recruitment, low pro-inflammatory cytokine production or high anti-inflammatory cytokine IL-10 secretion), lung edema and enhanced epithelial wound repair. On the other hand, phase I and phase II clinical trials showed safety results in patients undergoing chronic obstructive pulmonary disease [7], bronchopulmonary dysplasia [55] or idiopathic pulmonary fibrosis [56]. 

The first clinical trial was conducted in China. It was a phase I, single-center, randomized, double-blind, placebo-controlled clinical trial, in which ARDS patients received an intravenous injection of allogenic AT-MSCs (1 × 10^6^ cells/kg). The administered MSC seemed to be safe and well-tolerated in the patients [57]. A second phase I clinical trial was conducted in the USA, which included patients with moderate–severe ARDS who received a single intravenous injection of allogeneic BM-MSC (1 × 10^6^ cells/kg, 5 × 10^6^ cells/kg and 10 × 10^6^ cells/kg). In this multicenter, open-label and dose-escalation clinical trial, all the infunded levels were well tolerated. In a Swedish case report, two patients with severe, refractory ARDS received an intravenous infusion of allogeneic BM-MSC (2 × 10^6^ cells/kg) as a compassionate use. Both patients showed improvement with the resolution of respiratory, hemodynamic and multiorgan failure. In addition, a decrease in the biomarkers related to inflammation was also found [58]. 

In a phase IIa clinical trial conducted in the USA, in which patients received a high dose level of allogeneic BM-MSC (10 × 10^6^ cells/kg), no respiratory adverse effects were observed, and patients showed an improvement in the oxygenation index and a reduced level of ANG-2 in plasma, indicating that the MSC administration attenuated endothelial injury [59]. In addition, it was recently reported that the administration of menstrual blood-derived MSCs could reduce the mortality in patients with H7N9 virus-induced ARDS, without adverse effects after a five-year follow-up period in a clinical study conduce in China [60]. 

However, some issues were noticed regarding the MSC therapy for ARDS patients. Indeed, a higher mortality in ARDS patients treated with MSC was reported, but probably due to the more severe baseline illness in this treatment group [59]. Additionally, for lung function, one study found no significant change in a short-term (3–5 days) evaluation with MSC therapy [61].

To date, 17 July 2021, 11 clinical trials assessing the safety and efficacy of MSC therapy and its derived products therapy in ARDS patients have been registered in the open clinical trials database https://clinicaltrials.gov. Two of them have been completed and one resulted in safety and clinical improvement outcomes (Table 1). 

## 6. COVID-19: Facts and Challenges

### 6.1. Epidemiology

COVID-19 is a severe acute respiratory illness caused by SARS-CoV-2, which represents a devastating growing pandemic. Originating in Wuhan, China, by the end of 2019, COVID-19 moved to multiple other countries across the world. The World Health Organization (WHO) defined COVID-19 as a pandemic on 11 March 2020. The clinical profile of COVID-19 ranges from asymptomatic or pauci-symptomatic forms to respiratory compromission that requires invasive ventilation (14% of cases) and causes multiorgan failure (5% of cases).

The mortality rate fluctuates from 0.7 to 1.5–2%. To the date of 17 July 2021, there were 189,627,120 confirmed cases and 4,077,463 deaths because of this disease in the world. These figures are available from https://coronavirus.jhu.edu/map.html. Therefore, COVID-19 demands urgently new therapeutic perspectives.

### 6.2. Pathogenesis

Viral infections occur mainly due to aerosol particles, in which the virus survives up 3 h, due to droplet emission by the infected person when speaking or breathing, which pose an infection-threat if they are transferred to a close person. SARS-CoV-2 may pass through mucous membranes, particularly the nasal and larynx mucosa, and then enters the lungs through the respiratory tract. For cell entry, SARS-CoV-2 uses the ACE2 receptor and the transmembrane protease TMPRSS2 by its spike glycoprotein (SARS-CoV-2S), which is a class I fusion protein [62], both located on the alveolar type II pneumocytes and on the endothelial capillary barrier of the lungs. In particular, type II alveolar cylinder-shaped cells, representing 5% of all pneumocytes [63], are responsible for the generation of alveolar surfactant, act as “stem” and progenitor cells of type I pneumocytes (95% of pneumocytes) and perform gas exchange in the lungs. This explains, in part, that COVID-19 especially affects the lungs, which are the primary organ exposed to the virus due to its way of transmission, and which have a very slow regeneration rate. In addition, ACE2 is expressed in other organs, including capillary endothelial cells, smooth muscle cells, the heart, kidneys and digestive organs. This may explain why some patients with COVID-19 suffer from multiple organic dysfunction syndromes, such as thrombosis and coagulopathy, ARDS, acute myocardial damage, arrhythmia, acute kidney injury and shock [9].

The internalization of the virus in tissue and immune cells leads to activation of the NF-κB pathway and the secretion of inflammatory factors, such as TNF-α, INF-γ, IL-1 and IL-17 [64]. In most patients, the activated immune cells efficiently eliminate the virus, leaving the patient asymptomatic or manifesting a fever, dry cough, fatigue, myalgia and mild lung inflammation. However, in some patients, COVID-19 can progress to ARDS with significant hypoxia, multi-organ involvement and collapse that precipitates death, as well as high incidence of venous thromboembolic complications and abnormal coagulation parameters [65]. This is due to SARS-CoV-2 overstimulating immune cells, eliciting a strong immune response in the lungs. In this sense, COVID-19 represents a dramatic example of defense mechanisms overplaying in the presence of noxious agents, causing an inflammatory hyper-response that becomes more detrimental that the original aggression, due to its overwhelming action on tissues. In this sense, the over-activated pulmonary macrophages and the resulting inflammatory “cytokine storm” have been recognized as one of the determinants of the fatal consequences of coronavirus infection. The term “cytokine storm” has gained attention since it was first introduced in the 1990s to describe the onset of events aimed at modulating graft-versus-host disease (GVHD). Immune effector cells can release large amounts of pro-inflammatory cytokines (TNF-α and TGF-β, IFN-α, IFN-γ, IL-1β, IL-6, IL-12, IL-18 or IL-33) and chemokines (CCL2, CCL3, CCL5, CXCL8, CXCL9 or CXCL10) in response to SARS-CoV-2 infection [66] (Figure 1). After the “cytokine storm” at the local site, a massive influx of circulating granulocytes and monocytes occurs in the inflamed lungs, which subsequently results in pulmonary edema, remarkable proteinaceous exudate, pneumocytes hyperplasia, vascular congestion, air-exchange alteration, ARDS and secondary infection. 

In addition, “cytokine storm” spilling over to the systemic circulation and causing the weakening of the patient state has been observed to be the main predictor of mortality in COVID-19, and the identification and treatment of hyper-inflammation is recommended to reduce the rising mortality [67]. Several studies have also indicated that the probability of a worse prognosis of COVID-19 occurs in patients older than 60 years (more than 73% of total registered deaths), and that patients with comorbidities such as diabetes, high blood pressure or heart surgeries have a higher probability of getting infected. This may be due to a disturbed immune response development and/or a deficiency of virus elimination [68]. Thus, considering the increase in both age and co-morbidities, the enormous impact that this, and other possible future pandemics to hit our society, is reflected.

Moreover, there is a strong correlation between cytokine levels and the associated lung injury, ARDS and the poor prognosis in COVID-19 infection [69].

Notwithstanding, 1700 clinical studies have been registered worldwide, focused on investigating COVID-19 and trying to validate new therapeutic regimens, immunization protocols and anti-viral strategies. There are currently neither specific recommended antiviral treatments for treatment nor generalized vaccines readily available [70]. Antibacterial agents are ineffective due to the viral nature of the infection [71]. Thus, therapeutic schemes are limited to palliative care and assisted ventilation for patients with severe pneumonia. Systemic corticosteroids appear to be effective in severe COVID-19 patients. It has been reported that severe COVID-19 patients given 6 mg of dexamethasone once daily had 8–26% lower mortality compared to patients with standard care in a RECOVERY study [72]. However, corticosteroids have immunosuppressive effects, and there is a gap in the evidence regarding the long-term effect of dexamethasone, its repercussions in old-aged patients, the timing, the dose and even the type of administered corticosteroid. 

A range of accepted drugs for other diseases, such as remdesivir (Ebola), lopinavir–ritonavir (HIV), interferon 1β (multiple sclerosis) and chloroquine and hydroxychloroquine (malaria) are under investigation. For lopinavir–ritonavir and chloroquine and hydroxychloroquine, the FDA and the NIH cancelled its use due to undesirable data accumulation. In addition, the cytokines implicated in severe COVID-19 pathogenesis, such as IL-1 and IL-6, have been proposed to be potential therapeutic targets [73]. One of these drugs is tocilizumab, an immunoglobulin G1(IgG1) subclass antibody that inhibits the IL-6 receptor and that was tested in COVID-19 patients with elevated IL-6 levels, has exhibited several beneficial effects [74]. Recently, tocilizumab was added to remdesivir for hospitalized-patients with severe COVID-19-induced pneumonia in a REMDACTA study (NCT04409262). At present, the only available evidence-based COVID-19 treatment is remdesivir, a viral RNA-polymerase inhibitor, that shortens the time to hospital discharge. However, none of these drugs were able to prevent or attenuate the “cytokine storm” in the lungs from patients suffering with ARDS by SARS-CoV-2, whom usually need oxygen supply or mechanical ventilation [75].

The only exception could be fostamatinib, a spleen tyrosine kinase (SYK) inhibitor, known for its participation in a large number of immune pathways involving receptor signaling, cell adhesion, innate immune recognition and platelet function. Fostamatinib has been shown to play a role in the hyper-inflammatory response caused by anti-SARS-CoV-2-Spike IgG. 

In addition, fostamatinib is a potential mucin-1 (MUC1) antagonist, a transmembrane protein from mucosal epithelial cells, which takes an important part in the lining of the airway lumen. The overexpression of this protein by a subset of COVID-19 patients progressing to ALI and ARDS leads to excess mucus, which results in an increased duration of infection as well as mortality from respiratory illnesses. Given that, and the fact that it has been already FDA approved, fostamatinib was rushed into phase II clinical trials for COVID-19 [76].

Anyhow, a long-term, effective vaccine and a treatment for SARS-CoV-2 virus are far from complete. Despite the scientific community working tirelessly, many obstacles condition their mass production and stability at room temperature. Moreover, the clinical applicability of safe and effective antiviral agents remains a challenge. As a culmination, the rapid spreading of the pandemic requires viral infection and new solutions are needed to mitigate the impacts of this pandemic, especially in severe cases [77].

### 6.3. MSC-Based Therapy in COVID-19

In the middle of this stage, MSC, their conditioned medium and their EV strongly arise as possible therapeutic recourses, due to their anti-inflammatory, regenerative, pro-angiogenic, anti-fibrotic and antimicrobial capacities. In fact, MSC have been identified as an option that are able to control the “cytokine storm” and magnified immune response, as a result of their enhanced cytokine quality production. In addition, MSCs from different organs or donors have been reported to be negative for ACE2 and TMPRSS2, suggesting that MSC are free from COVID-19 infection [78].

According to the International Society for Stem Cell Research (ISSCR), there are no approved stem-cell-based therapies to treat COVID-19. Lately, due to the advances in cell-based therapy, stem cell and regenerative medicine have focused on MSCs as one of the therapeutic approaches for treating COVID-19 [79]. 

It was also suggested that the utility of MSC therapy is more evident in particularly severe COVID-19 cases. In addition, MSC have shown a high degree of tropism/homing to the lungs after their intravenous administration (the most common route for cell therapy application), which was considered a limitation for the treatment of various pathologies, but this represents an advantage for the treatment of acute lung diseases. Infused MSC were detected in the lungs in a high percentage of patients within 30 min, a phenomenon that is further increased by inflammation [80]. Interestingly, as noted above, these trapped cells in the lung are able to secrete a wide variety of soluble factors that efficiently ameliorate the lung injury. 

An early study showed that human UC-MSC therapy was safe and effective at modulating the immune response and repairing injured tissue in a critically ill 65-year-old patient affected by COVID-19 [81]. MSCs were administrated intravenously three times (50 × 10^6^ cells, each time, every 3 days). After the second UC-MSC administration, there was a clear recovery in the laboratory parameters and vital signs. Thereafter, the white blood cells and neutrophils counts decreased to a standard level, whereas the number of lymphocytes increased to its usual value. 

The second study involved seven patients with COVID-19-related pneumonia (one presenting a severe critical type, four exhibiting severe types and the remaining two exposing common types of the syndrome), who received an intravenous administration of 1 × 10^6^ BM-MSC per kilogram of body weight. A further three patients with severe types of COVID-19 were enrolled as controls. No apparent adverse effects were detected following the single MSC injection. The patients receiving MSCs showed improved clinical and analytical endpoints and also reduced viral titers 2–4 days after receiving the MSC infusion. Interestingly, this study by Leng et al. suggests a strong anti-inflammatory effect after an MSC infusion in COVID-19 patients [9]. This resulted in a decrease in the white blood cells and neutrophils counts, a 10-fold drop in C-reactive protein secretion, a cytokine-secreting immune cells decline and a diminution in proinflammatory cytokine TNF-α levels. In contrast, the number of peripheral lymphocytes, a group of regulatory dendritic cells and levels of the anti-inflammatory protein interleukin-10 augmented in peripheral blood. 

However, some concerns were raised regarding the preliminary MSC therapy for COVID-19 or ARDS patients. In this way, a retrospective study on the efficacy and side effects of MSC therapy in severe COVID-19 described a significantly high serum level of lactate, cardiac troponin T and creatine kinase-MB after its application, posing a risk for patients with metabolic acidosis or coronary heart disease [82]. 

Thus far, over 26 clinical trials involving MSC have been conducted to investigate their therapeutic safety/efficacy in patients with ARDS secondary to COVID-19 (https://clinicaltrials.gov, accessed on 17 July 2021). Two of them have been completed (Table 2), and most of them are nearing completion. The preliminary findings are promising, and no severe side effects have been reported during the treatment period. As regards the MSC-derived extracellular vesicles, a completed clinical trial is already available with results pending publication, as shown in Table 2.

Additionally, at the time of writing this manuscript, there are more than 70 MSC-based clinical trials for COVID-19 registered at the World Health Organization-International Clinical Trial Registry Platform (WHO-ICTRP) (https://www.who.int/clinical-trials-registry-platform, accessed on 17 July 2021) and on the NIH ClinicalTrials.gov website [78]. The largest number of ongoing clinical trials are in China (*n* = 32) and the USA (*n* = 18). However, the number of countries where MSC-based clinical trials are registered in patients incurring COVID-19 is progressively increasing, for example, Spain, France, Germany, Denmark, Sweden, Russia, Ukraine, Belarus, Canada, Brazil, Mexico, Turkey, Iran, Jordan, Pakistan or Indonesia. In general, these studies recruited patients with severe COVID-19 that were under invasive mechanical ventilation and receiving antiviral and/or anti-inflammatory treatments (including, but not limited to, steroids, lopinavir/ritonavir, hydroxychloroquine and/or tocilizumab) and showed no clinical improvement. In most of these studies, the patients were randomly divided into the following two groups: the standard COVID-19-treatment group and the MSC-based therapy plus the standard COVID-19-treatment group. Moreover, intravenous allogeneic cell-therapy has been applied with MSC derived mainly from UC, AT or BM, but also with MSC from other different human origins, such as WJ, olfactory mucosa, dental pulp, menstrual blood, decidual or placental [83,84]. At the same time, an intravenous injection of MSC ranging from 0.5 × 10^6^ to 750 × 10^6^ cells/kg is being used in these clinical trials. The number of injections varies between a single and up to three doses during short time intervals of 2 or 3 days. In addition, three clinical trials were based on MSC-derived exosomes, two of which used aerosols. 

Overall, the diverse MSC treatments were considered safe and well tolerated. Since clinical improvement was observed in the most serious COVID-19 cases, MSC therapy was also evaluated as effective in combination with conventional treatment regimes. An infusion of MSCs or MSC-derived exosomes enhanced immune indicators (including CD4^+^ T cells and other subtypes) and decreased inflammatory markers (interleukin-6 and C-reactive protein). Oxygen saturation (SaO_2_) and partial pressure of oxygen (PO_2_) recovered after MSCs infusion. Qu et al. published a systematic review and random-effects meta-analysis in humans to determine the potential value of MSC therapy for the treatment of COVID-19-infected patients with ARDS [85]. Nine studies were included in this meta-analysis and the combined number of patients was 117. The MSC used were allogeneic human cells derived from BM, UC, AT, menstrual blood, or undeclared sources. No severe adverse events were observed, and minor harmful reactions resolved naturally. The authors concluded a positive trend in combined mortality after the use of MSCs, but a statistical significance was not achieved. Likewise, an improvement fashion was found in radiographic findings, pulmonary function and the levels of inflammatory biomarkers. No comparisons were made between MSC from different sources.

With regard to MSC-derived products, in a preliminary study, single intravenous injections of BM-MSC-derived exosomes were given to the 24 COVID-19 patients who met the criteria for moderate-to-severe ARDS. The infusions of these exosomes were well tolerated, significantly improved lung function, alleviated systemic inflammation and increased the total number of circulating neutrophils and lymphocytes in a majority of the COVID-19 patients. In addition, exosomes efficiently attenuated SARS-CoV-2-related ARDS in 71% of treated patients [86].

According to disponible data, most clinical trials are in the initial phases (phase I, I/II, II). However, it is of note to say that there is one registered phase III clinical trial that uses a cell product named Remestemcel-L for the treatment of patients with moderate and severe ARDS caused by COVID-19. Remestemcel-L is a third-party, off-the-shelf suspension of ex vivo cultured adult human MSC intended for intravenous infusion. It was approved for use in Canada in May 2012 under the trade name Prochymal^®^ by Osiris Therapeutics, for the management of refractory acute GvHD in children who are unresponsive to systemic steroid therapies [87]. 

The human and economic losses caused by the pandemic have speeded the search for new effective and secure therapies against COVID-19. In this sense, the differentiation capacity and the paracrine activity of human amniotic membrane stem cells (hAMSC) capable of allowing a secretion decrease in pro-inflammatory cytokines at the lung injury focus and suppressing the exacerbation of the immune system in ARDS caused by SARS-CoV-2 virus, would probably lead to a mitigation of cytokine storm and repairing and replacing the damaged tissue. However, it is necessary to develop preclinical studies and clinical trials with proven efficacy, especially in coronavirus-induced respiratory diseases [78]. 

Despite all these positive data obtained from clinical trials, there are doubts about the application safety of this type of incipient therapies in ARDS, such as what should be the origin of the MSC, the doses, routes and frequency of their administration (regimen single versus multiple doses) [88]. In addition, it was strongly argued that MSC should not be infused during the early period, where inflammation is vitally important and beneficial to contain viral infection. In this context, if misused, MSC could act as a double-edged sword, as too much immunosuppression can abolish the “physiological inflammation” necessary to control viral replication [77].

## 7. Restrictions of MSC-Based Therapy

Generally, there are positive results from the clinical trials based on MSC therapy, but there are also some discordant outcomes. For example, despite the initial encouraging results, a recent review of completed randomized clinical trials using MSC for the treatment of GvHD found that MSC might have little or no effect. 

The disparities found in the effectiveness of the use of MSC in clinical studies may be due to the whole quality of the study design, tissular origin of MSC, tissue processing, donor sex, age, medical history, differences associated with MSC manufacture under harvest conditions, reductions in cell quality during in vitro expansion, administration routes, doses and dosing intervals, modest cell survival after in vivo transplantation or incompetent homing capacity to targeted sites. In addition, an excessive inflammatory immune response, oxidative stress or hypoxic microenvironments at the injury locations are also factors restraining the MSC’s survival and engraftment [89,90].

Although MSC have the same identifying molecular markers, they differ functionally in terms of proliferation capacity, transdifferentiation or immunophenotype depending on their tissue origin, by both paracrine and microvesicle mechanisms through secretome-derived products [90,91]. Thus, for example, a recent meta-analysis suggested that BM-MSC and UC-MSC showed better therapeutic results than AT-MSC in preclinical studies of ALI. Donor age is a significant factor affecting MSC’s efficacy. The MSCcultivated from neonatal tissues exhibit a longer active life, higher proliferation rate and differentiation potential when compared to adult tissues. Furthermore, the MSC derived from unhealthy donors may exhibit negative clinical outcomes when used for therapies [92]. Other important aspects to take into account are the components of the culture media that may affect cell phenotype, such as the damage caused by cryopreservation and subsequent thawing and oxygen concentration. High oxygen levels may compromise the therapeutic benefits of MSC. Native MSC tissue environments range between 1 and 7% O_2_. During culture, cells sense an oxygen concentration of 20%, which may cause oxidative stress affecting the viability, and eventually, senescence.

On the other hand, there are several issues related to cell therapy, such as several safety considerations potentially associated with transplanting live and proliferative cell populations, including immune compatibility, tumorigenicity, occlusion in microvasculature [17] and infectious transmission. All this may aggravate the prognosis due to the abnormal clotting parameters reported in many critically ill patients with COVID-19 [93].

It should be noted that, in an ex vivo model of ARDS, intravenously administered MSC adhered to membrane oxygenator fibers during extracorporeal membrane oxygenation support, resulting in a rapid decline in oxygenator performance through the circuit. In addition, it has been suggested that the intravascular administration of MSCs in susceptible patients could cause a transient increase in pulmonary pressures and pulmonary edema [94].

An equally important restriction is the economic cost of these therapies. A regular stem cell therapy was estimated from USD 4000–8000 in the United States, and the cost for enlarged harvested cells ranges from USD 15,000–30,000. Therefore, more research is needed to assess the cost-effectiveness of cell-based therapy and ensure sustainable access to patients and the general population [95].

## 8. MSC-Derived Products as Cell-Free Therapy and Their Restraints

The primary concept of MSC therapy was based on its ability to autotarget injury sites and differentiate into diverse cell types contributing to tissue regeneration. However, several studies have found that the implantation time of MSC is generally too short for an effective impact. In fact, <1% of MSC were said to survive more than a week following systemic administration, and their contribution to new tissue creation is usually minor [96]. Accumulated experience discloses that the beneficial effects of MSC are mainly caused by their secretion of soluble paracrine factors. These include proteins (growth factors and cytokines) and extracellular vesicles (EV). Due to the regenerative, anti-inflammatory, anti-oxidative stress, angiogenic and anti-apoptic power of these biological products, MSC secretome may be considered a successful aspirant for a direct medical biotechnology [97]. This strategy avoids issues arising from the use of stem cells themselves, amongst them, the following stand out: (i) the uncertainty related to the transplantation of proliferating living cell is resolved, including immunological incompatibility, tumorigenicity, emboli formation, transmission of infections and potential entry of MSC into senility; (ii) unlike cell therapies, secretome can be better assessed in terms of their safety, dose and potency, such as conventional therapeutic agents; (iii) secretome can be stored without requiring potentially toxic cryopreservative vehicles; and (iv) the use of secretome-derived products, such as the conditioned medium or exosomes, is both more cost-effective and convenient for clinical use, since the use of secretome may avoid the time and costs associated with expanding and sustaining clonal cell lines. This has to do with the fact that secretome for therapies might be prepared in advance in large amounts and be available for treatments when necessary [17]. 

Background studies suggest the safety and efficacy of the MSC-derived secretome is suitable for topical, intravenous and oral administration [98]. Nevertheless, the secretome as a whole may have the limitation of being an overly biologically complex product, which can make the identification of a vigorous mechanistic explanation difficult for its therapeutic effects. Thus, a possible substitute might be to use a more specific part of its components, such as EV. In support of this proposal, it should be noted that while MSC are damaged by shear stress and retained in the pulmonary vasculature, EV are not subject to these processes [99]. 

EV represent a major component of the MSC secretome that have inspired an extraordinary interest as a burgeoning option to MSC-owned activities. EV can be categorized as (i) exosomes (30–120 nm in diameter), which come from the cell in endosomal compartments known as multivesicular bodies; (ii) microparticles (150–1000 nm in diameter), caused by blisters emerging from the plasma membrane and then liberated after the cytoskeletal proteolytic cleavage; and (iii) apoptotic bodies (500–2000 nm in diameter), released during the programmed cell death process. Among these microparticles, exosomes excel by their functional significance. Their biogenesis is a process composed of the following four phases: initiation as endosomes, endocytosis, multivesicular bodies and release. Exosomes are membrane-attached phospholipid particles secreted by cells. They comprise a wide array of different components, such as RNA, lipids, proteins, cytokines, chemokines and interleukins, integrins (CD81, CD63 and CD9), transport proteins (annexins and Rab GTPases), signal transduction factors (kinases), cytoskeletal proteins and metabolic enzymes. Through the horizontal transfer of all these biologically active factors, exosomes portray an important intercellular communication route in organ homeostasis and organ disease. These mechanisms include binding to surface receptors to activate signal cascades, the internalization of surface-bound exosomes and fusion with the cell to transport material directly to the cytoplasmic membrane and cytosol. 

Experimental studies reveal the emerging role of these products in a range of pathologies [18]. In line with ARDS, several studies found that the administration of MSC-derived MV by inhalation or an intravascular route was as effective as MSC application through different mechanisms. MSC-derived EV have been shown to reduce inflammation and alveolar edema by decreasing the inflow of inflammatory cells and restoring alveolar–capillary permeability after endotoxin lung injury. The involved mechanism corresponds to the FGF7 mRNA or angiogenin-1 transferring, both of which have anti-inflammatory and reparative properties in lung tissues [100].

In addition, exosomes show substantial benefits for their application in therapies: they are smaller, less complex and less immunogenic than their ancestor cells (since they contain lower levels of membrane-bound proteins) and their production and stockpiling are also simpler than for such progenitor cells. Moreover, other merits of exosomes include a longer half-life in blood circulation, tropism specifically targeting inflamed tissues and tumors cells, as well as the faculty to pass through the hematoencephalic barrier [101]. 

This latter aspect may be of potential importance to treat COVID-19 patients, since neurological disorders, such as severe cerebrovascular diseases, altered consciousness and loss of smell or taste have been described as forms of presentation of the disease.

In contrast, the cell release and exosomes transfer capacity can be augmented through a variety of strategies, such as prolonged culture and holding cells at an acidic pH or a low O_2_ pressure [102]. Preclinical studies have demonstrated the safety of MSC-derived exosomes and the possibility of large-scale production at clinically relevant doses. 

To date, four clinical trials involving the therapeutic application of exosomes in patients suffering from ARDS in the COVID-19 context have been initiated, and only one of them has been completed, as indicated in Table 2. This information can be found at www.clinicaltrials.gov (late accessed date: 17 July 2021).

As research on the potential of exosomes for COVID 19 treatment has shown, the strategies for their use can be divided into the following three general categories: using exosomal particles secreted by different sources of MSC instead of cell therapy, incorporating specific miRNAs and mRNAs into exosomes and handling exosomes as promising nanocarriers to deliver drugs to treat COVID-19. With regard to this latter active-load therapeutic platform possibility, it was recently reported that microvesicles derived from human WJ-MSC enhance autophagy and ameliorate lung injury via the delivery of miRNA-100 [103]. It was also proposed that the immunomodulatory cargo of MSC exosomes combined with the antiviral drugs make them a new intervention tool for treating the disease [104]. In this sense, the chance of charging exosomes with the antiviral drug remdesivir, prescribed for the treatment of patients with COVID-19, was also suggested [105]. 

There is limited information on the potential risks of therapy based on secretome products. These potential risks, likely related to the administration of exogenous biological products, appear to be small compared with cell-based therapies. 

Presumably, the safety concerns may be related to the immunogenicity and immunosuppressant properties of the MSC secretome. Although secretome is known to contain potentially immunogenic EV, this feasible immunogenicity was found to be lower than that of their parent MSC [106]. On the other hand, considering that MSC secretome has immunosuppressive attributes (since it has been shown to constitute one of the main mechanisms of action in the treatment of autoimmune diseases [107,108]), its use might tentatively increase the risk of infection, immunodeficiency and tumor growth when treating patients. In addition, it has been signaled that the heterogeneity of EV due to several factors such as their source, isolation, purification techniques (centrifugation, ultracentrifugation or size exclusion chromatography) and aging, may result in functional differences. Thus, while some EV could have both beneficial immunomodulatory and regenerative activities, others could contribute to the acceleration of infection by transporting viruses within them and bypassing the immune system [109]. On the other hand, an experimental study reported that the effects of MSC and EV differ according to ARDS physiopathology [110].

As mentioned earlier, some data evidence that the immunoregulatory effects of MSC in vitro depend on the condition of the inflammatory microenvironment. Although this status could also exist under in vivo conditions, it will be necessary to define, in the light of clinical circumstances and therapeutic interest, the optimum amount of secretome to achieve the right balance between safety and effectiveness.

In terms of production limitations, there are two technical issues, the sources and the instability of the secretome, which need to be addressed. For example, the amount of MSCs required to produce enough secretome for an equivalent effect on acute wounds is about 10–25 times higher than the live cells administered directly [111]. These very high demands in production enhances the manufacturing, quality control derivation and validation costs because the function of these cells can change with repeated passages. Moreover, the problem that challenges the secretome therapy can be its possible instability and lack of potency. However, there are ways of dealing with all these limitations. These include optimal and productive methods of cell culture expansion that enhance the functional capability of MSC and their secretome derivatives.

## 9. Perspectives on MSC and Their Derived Secretome Production for Future Therapies

Based on all of the above about the impending new demands for MSC-based therapies and the current limitations on their in vitro production, we are confronted with the need to implement innovating strategies that increase and enhance their in vitro expansion and MSC-derived secretome production. How far MSC exosomes can be used for cell-free regenerative medicine will depend on the optimal choice of the ideal MSC, dictated by the tissue origin and donor parameters for each application, the establishment of appropriate standards for in vitro MSC expansion and the evaluation of functional potency tests of the obtained products. (Figure 2).

### 9.1. The Optimal Cell for Every Request

Bearing in mind the functional disparities of MSC and the factors that impact it, not only due to the type of donor, but also because of their origin from different bioniches, the most appropriate MSC should be selected for each indication in years to come.

Patients suffering from systemic maladies such as diabetes, obesity, systemic lupus erythematosus and rheumatoid arthritis are known to have functional impairments in their MSC, which must be considered in order not to see them as ideal donors. Furthermore, the aging of MSC during expansion is a crucial limiting factor because older cells lose the skill to behave as stem cells and tend to be prone to senescence, even to transform.

Aging MSC are more likely to turn on a senescence-associated secretory phenotype and to produce pro-inflammatory cytokines such as IL-1, IL-6 and IL-8, which suppress the regenerative process [112]. It is also known that culture expanded MSCs, in general, gradually lose their self-renewing and multipotency abilities [113,114], which represent a major limitation for investigating potential treatments [34,115]. It was also revealed that MSC are unable to perform more than 30 to 40 population doublings, while immortalized MSC are in position to perform more than 200 population doublings [116,117,118]. Immortalization, which requires the silencing of tumor suppressors p53- and Rb-mediated pathways and telomere preservation, helps these cells to increase their proliferation rate and prevent senility, whilst retaining both mesenchymal phenotype and multipotency. MSC immortalization can be achieved by combining the transduction of two immortalizing oncogenes, such as simian virus 40 large T antigen (SV40LT), which promotes cell cycle progression, and human telomerase reverse transcriptase (hTERT), which inhibits telomeres shortening.

On the other hand, it is known that MSC functionally differ regarding their tissular origin. Thus, for example, the distinctions between AT-MSC and BM-MSCs were reported with respect to proliferation, differentiation or paracrine mechanisms, such as the secretion of pro-angiogenic molecules, namely, the extracellular components of MMP [91]. In addition, we have to consider that MSC and their secretome-derived products of another alternative origin have not yet been used in ARDS experimental models. This is the case of human uterine cervical stem cells (hUCESC), which showed a high proliferative rate in pre-clinical studies, and whose secretome has formidable regenerative, anti-inflammatory and anti-microbial capacities [119,120]. This special type of MSC, which was found in the transformation zone of the human uterine cervix, is an example of the importance of the MSC´s origin tissue, since their location in this organ might display a protective effect against the penetration of potentially harmful elements, such as some strains of the human papillomavirus family. Consequently, it has been purposed that the immunoregulatory capacity of its secretome may be of interest against the inflammatory mechanism associated with ARDS, including COVID-19. 

Genetic manipulation of MSC using the application of defective viral replication vectors, such as lenti- and adenoviruses, offers the possibility to strengthen some of their capabilities. Thus, there is evidence advocating that the incorporation of anti-inflammatory genes to MSC (for example, IL-10, HGF, IDO or Foxp3) could strengthen their therapeutic aptitude. In the same way, over-expression induced by a number of factors has been reported to improve apoptotic tolerance, cell survival and more angiogenic, neuroprotective, osteogenesis or anti-cancer interventions [91]. Nevertheless, despite all these positive data on the genetic manipulation of MSC, their clinical introduction has several limitations. The application of the aforementioned replication-faulty viral vectors is associated with toxicity, immunogenicity and potential tumorigenicity [121]. Most recently, many studies have shown that the Clustered Regularly Interspaced Short Palindromic Repeats (CRISPR)-Cas system may increase the therapeutic potential of MSC by attacking and degrading foreign DNA, allowing its integration into the chromosome [122]. For instance, the use of engineered BM-MSC expressing IL-10 excessively using CRISPR activation has been reported to treat myocardial infarction in an experimental design [123].

### 9.2. MSC In Vitro Production

Although MSC are widely distributed in most organs and tissues, they are found in trace amounts. For example, MSC isolated from BM occupy only about 0.001–0.01% of mononuclear cells [124], 0.3% from UC [125], or 1.2% from human AT, in healthy adults [126]. These quantities of MSC are far beyond those required in clinical applications, which are around 2 × 10^6^ cells/kg of body weight per dose [127]. In addition, for some patients and diseases, repeated administrations of MSC up to several hundred million are required to achieve the effective therapeutic effect [128]. Consequently, the MSC in vitro expansion is mandatory for many weeks before completing enough cells for cell-based therapies. 

Most centers use the traditional production system in T-flasks. Nonetheless, this kind of method is only practical to treat a small number of patients [128]. Thereby, there is an exponential increase in the establishment of an efficient large-scale expansion technique to obtain the largest number of cells claimed, in a short time frame and on a cost-effective manner, without compromising cell quality. All of them are preconditions for meeting the demand of late-phase clinical trials and future commercialization. 

There are several bioprocessing strategies for broad production, such as multilayered flask, spinner flask, roller bottle or biorreactor, used for the expansion of different sources of MSC (AT-MSC), UC-MSC, WJ-MSC, BM-MSC, periosteum-derived MSC, villous chorion-derived MSC, dental pulp-derived MSC and fetal MSC. In line with the fact that the production of adherent MSCs depends on the surface area, it is key to achieving a maximum covering area in which the original biological characteristics of the phenotype and the potency of the MSC will be preserved. A multilayered flask is a specially designed culture flask that consists of multiple layers with a large area for cell culture. A spinner flask and roller bottle are dynamic culture systems that create a shear stress to cells as it involves the mechanical shaking of the culture medium or culture vessel to provide a more efficient nutrient transfer. However, all of these systems are manual bioprocess strategies with lesser efficiency [129]. Alternately, properly controlled automated bioreactors ensure efficient mixing in a closed system for extended-scope expansion in lot size with reduced labor and time [130]. The growth of MSC in bioreactors enables us to advance in product quality on a number of fronts, including commercial manufacturing, greater traceability through monitoring and surveillance, the ability to eliminate errors and operator-related contamination, the avoidance of batch variability, and, finally, the steep fluctuations in pH, oxygen concentration or nutrient gradients induced by manual medium exchange. Stirred tank reactors are the most commonly used devices for massive MSC expansion. There are some key technological aspects to consider in this more advanced type of cell culture expansion system, such as the use of microcarriers, hydrodynamic parameters and agitation. Microcarriers are 100–300 μm diameter structures made of various materials, such as polystyrene, dextran, cellulose, gelatin, glass or decellularized tissue, with different surface characteristics [131]. These systems tolerate the growth of cultures in 3D [129,132], which offers many benefits in terms of optimal MSC production and its greater regenerative, angiogenic, anti-inflammatory, immunomodulatory or anti-apoptotic potency [132,133]. 

A better understanding of the bioprocess parameters that influence the MSC´s therapeutic efficacy remains, therefore, essential. In this respect, there are many possible ex vivo MSC changes to optimize the bioreactor conditions and, therefore, maximize the amount of MSC without relinquishing quality and therapeutic potency [134].

### 9.3. Ex Vivo MSC Modifications: Close to More Specific Therapeutic Applications

Ex vivo adjustments that can be made to reinforce the therapeutic interest of MSC in cultures include oxygen or pre-conditioning with inflammatory cytokines.

Evidence proves that MSC grown at low oxygen concentrations enhance several therapeutic effects by expressing higher levels of pluripotent and proliferation markers [135,136], increasing cytokine secretion and growth factors within transplanted stem cells, stimulating angiogenesis [137], migration to injury sites and anticancer effects [102,138].

Another pre-conditioning policy to improve the MSC´s therapeutic benefits involves exposure to an inflammatory environment where inflammatory cytokines are present, such as IFN-γ [139] and TNF-α [140]. Treating MSC with these inflammatory cytokines or their combinations increases their secretion of anti-inflammatory biomolecules and amends their immunosuppressive function [91]. With respect to ALI, Park et al. observed an increase in antimicrobial activity in an ex vivo perfused human lung injured with severe Escherichia coli pneumonia when MSC were pretreated with the TLR-3 agonist. Similarly, Song et al. showed that pretreatment with IL-1β increased the immunomodulating effects of MSC, partially by an exosome-mediated transfer of miR-146a.

These data indicate the potential to modulate the capacity of MSC and their secretome, based on different chemical or molecular stimuli. In this way, we could envisage the opportunity in the future to adapt the potentiality of MSC secretome to the optimal therapeutic application required for any specific condition.

### 9.4. Towards Standard Functional Tests for Each Therapeutic Application

In the present situation of allogenic therapies, MSC produced from one or more selective donors are used as a universal medication for manifold patients. However, many studies have noted the importance of both donor-to-donor and tissue source variations [141,142]. Furthermore, MSC are isolated from several different tissue source materials and as mentioned above, there is a functional miscellaneous with regard to their origin. On the other hand, MSC are produced with different cultivation or pretreatment strategies [143]. Although only a few studies investigated the influence of bioprocess parameters on MSC therapeutic potency [132,144], it was reported, for example, that variable in vitro expansion strategies have a greater impact on MSC’s molecular phenotype than donor age [145]. Therefore, to minimize inter-donor variations and reduce bioprocess variability, large-scale MSC production for allogeneic therapies is needed straightaway. This clearly underlines the importance of unifying inter-laboratory practices for the manufacture of MSC and their secretome derivatives, in order to attain excellence in the biomedical application of cell-based therapy. However, no admissible potency tests exist for MSC to release for clinical therapies that predict their in vivo efficacy. Therefore, the optimal functional characterization of the manufactured MSC and their secretome should also be performed, especially for each specific therapeutic indication.

## 10. Conclusions 

ARDS represents a real current challenge for medicine due to its incidence, the tremendous morbidity and mortality it generates and the absence of an optimal treatment that globally tackles the multiple nuances of this complex physiopathological process. 

The COVID-19 outbreak only increased the urgent demand for an affordable, safe and effective treatment for this process. Preclinical studies and early clinical trials suggest the therapeutic usefulness of MSC and their derivatives for these processes. Preclinical studies and early clinical trials suggest the therapeutic utility of MSC and their derivatives in acute lung injury (ALI) and ARDS.

Those cellular or biological therapies show antimicrobial, anti-inflammatory, regenerative, angiogenic, antifibrotic, anti-oxidative stress and antiapoptotic actions, which can thwart the kaleidoscope of physiopathological mechanisms engaged in ARDS. It is also relevant to consider the opportunity of using MSC secretome-derived products, such as extracellular vesicles and, specially, exosomes, which may reproduce the therapeutic effects of MSC in lung injury.

The use of conditioned medium or EV can dodge the safety concerns associated with MSC administration, and they may also be given in different formulations. Thus, MSC secretome was suggested to be formulated in an inhalable and injectable form, persisting stable in the blood until delivery into the lungs. Furthermore, considering the treatment of a pandemic, the costs of MSC secretome likely appear to be higher than those of monoclonal antibody therapy. However, the following different limitations must be addressed: (i) the selection of the optimal MSC, bearing in mind both the biological variability within donors and across different biological niches, (ii) the massive obtention of MSC secretome derivatives, possibly by immortalization or other genetic engineering of the most suitable MSC, (iii) the use of bioreactors that allow its growth in 3D, (iv) optimal culture conditions (such as O_2_ tension, type of medium and supplements, substrates and extracellular cues, inflammatory stimuli, etc.) and (v) use of the adequate functional testing of these biological products before medical application. Moreover, incorporating artificial intelligence tools can contribute to the successful integration of all this new budding evidence.

Naturally, therapeutic applications should be conducted in the context of well-designed, case–controlled clinical trials framed by a strict, ethical and highly regulated production process, with the permission of an appropriate authority, with the aim of obtaining mechanistic information as far as is practicable. However, urgent and well-managed actions must also be initiated, considering the unsolved ARDS, the irruption of a highly transmissible virus, SARS-CoV-2, that spread in only 10 months and infected more than 40 million people in 214 countries worldwide, and other very probable pandemics to come.

## Figures and Tables

**Figure 1 ijms-22-07850-f001:**
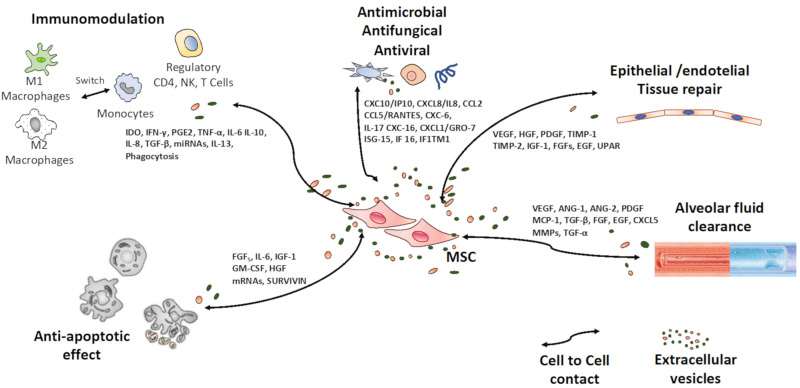
Known and probable therapeutic effects of MSCs on Acute Respiratory Distress Syndrome. Mesenchymal stem cells (MSCs) playing a key role as “traffic controllers” of intercellular signals in different physiological processes through secreted factors (secretome) or by cell-to-cell contact.

**Figure 2 ijms-22-07850-f002:**
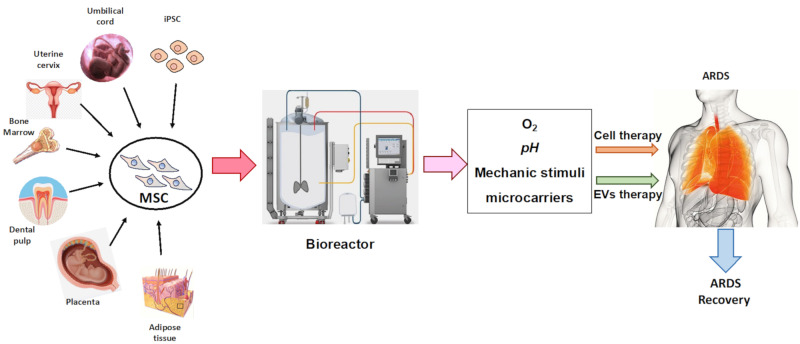
Mesenchymal stem cells (MSC) production strategies based on the source of MSC, the system used for in vitro expansion, culture cell conditions and the ultimate product for Acute respiratory distress syndrome (ARDS) therapy.

**Table 1 ijms-22-07850-t001:** Completed clinical trials assessing the safety and efficacy of MSC therapy in ARDS.

CT Title	Trial Phase	MSC Source	Patient Status	N	Route of Administration	Endpoints	Dose	Results	Reference n°
**Mesenchymal Stem Cells (MSC) for Treatment of Acute Respiratory Distress Syndrome (ARDS) in Patients with Malignancies**	1	Allogeneic BM-MSC	Blood and marrow transplantation ARDS	20	Intravenous	Adverse events Clinical improvement	3 × 10^6^ cell/Kg	No study results posted	NCT02804945
**Human Mesenchymal Stem Cells for Acute Respiratory Distress Syndrome (START)**	1	Allogeneic BM-MSC	ARDS	9	Intravenous	Adverse events Ventilator and ICU free days to day 28 Hospital survival to day 60 Mortality at hospital discharge	1, 5 and 10 × 10^6^ cells/kg predicted body weight (PBW)	Intravenous infusion of allogeneic, BM-MSC was well tolerated in patients with ARDS	NCT01775774

**Table 2 ijms-22-07850-t002:** Completed clinical trials assessing the safety and efficacy of MSC and MSC-derived extracellular vesicles therapy in ARDS secondary to COVID-19.

CT Title	Trial Phase	MSC Source	Patient Status	N	Route of Administration	Endpoints	Dose	Results	Reference n°
**Cellular Immuno-Therapy for COVID-19 Acute Respiratory Distress Syndrome**	1–2	Allogeneic UC-MSC	COVID-19 ARDS	12	Intravenous	Adverse events Numbers of patients alive to day 28 Ventilator free days to day 28	25, 50 and up to 90 × 10^6^ cell/unit dose	No study results posted	NCT04400032
**Use of UC-MSC for COVID-19 Patients**	1–2	Allogeneic UC-MSC	COVID-19 ARDS	24	Intravenous	Adverse events Survival rate after 90 days post first infusion Ventilator free days to day 28 Clinical Improvement	100 × 10^6^ cells/infusion	No study results posted	NCT04355728
**Extracellular Vesicle Infusion Treatment for COVID-19 Associated ARDS (EXIT-COVID19)**	2	Allogeneic BM-MSC	COVID-19 ARDS	120	Intravenous	Adverse Events All-cause mortality to day 61 Pa02/Fi02 ratio to day 61 Clinical, laboratory parameters and quality of life Improvement	800 billion and 1,2 trillion EV	No results posted	NCT04493242

## Data Availability

Not applicable.

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
