# Peer review of "Mesenchymal Stem Cell-Based Therapy as an Alternative to the Treatment of Acute Respiratory Distress Syndrome: Current Evidence and Future Perspectives"

_ijms, 2021, doi:10.3390/ijms22157850_

Round 1
Reviewer 1 Report
The article -
"Therapeutic mechanisms of mesenchymal stem cells in acute respiratory distress syndrome reveal potentials for Covid-19 treatment" presented a table - "Clinical trials: Registered MSC-based treatment in ARDS." This table showed several Covid-19 trails with MSC. Please reflect upon them and add further updates on the trails.
Author Response
Dr. Francisco Vizoso
Unidad de Investigación
Fundación Hospital de Jove
Avda. Eduardo Castro, 161
33290 Gijón, SPAIN
Fax: +34 985103564
E-mail: investigacion@hospitaldejove.com
July 17th, 2021
Dear Editor,
Please find enclosed the revised version of our manuscript (ijms-1312284) where we have responded to all the reviewer's comments. We thank you and Reviewers for the valuable criticisms and suggestions, which will undoubtedly translate into an improved version of the review.
Our specific answers to the reviewer’s comments are listed below.
Reviewer 1
We would like to thank the examiner for the very positive feedback on our work. This certainly helps us to pursue our work with more support and confidence. The clarity in your comments helped to enhance our paper.
"Therapeutic mechanisms of mesenchymal stem cells in acute respiratory distress syndrome reveal potentials for Covid-19 treatment" presented a table - "Clinical trials: Registered MSC-based treatment in ARDS." This table showed several Covid-19 trails with MSC. Please reflect upon them and add further updates on the trails.
According to reviewer suggestion, we have updated the information concerning the clinical trials recorded to date on MSC-based treatment in ARDS in no COVID-19 and COVID-19 patients. This led us to add a completed clinical trial on MSC-derived extracellular vesicles therapy in ARDS secondary to COVID-19 in table 2, and to perform subsequent changes to the text, as we indicate below:
Section 5, page 8, line 381: “To date 17 July 2021, 11 clinical trials assessing the safety and efficacy of MSC therapy and their derived products in ARDS patients have been registered in the open clinical trials database https://clinicaltrials.gov”.
Subsection 6.3., page 12, line 563: “As regards MSC-derived extracellular vesicles, a completed clinical trial is already available with results pending publication, as shown in Table 2”.
Subsection 6.3., page 12, line 566: Figure title: “Completed clinical trials assessing the safety and efficacy of MSC and MSC-derived extracellular vesicles therapy in ARDS secondary to COVID-19”.
Section 8., page 16, line 754: “To date, 4 clinical trials involving the therapeutic application of exosomes in patients suffering from ARDS in COVID-19 context have been initiated, and only one of them has been completed, as indicated in table 2”.
Reviewer 2 Report
Page 2
Line 57
in vivo should be set in italics.
Line 69
Please correct word Resucitation as Resuscitation
Page 3
Line 127
Please add reference for paper by Friedenstein et al.
Page 4
Line 152
If you start the paragraph with "it has been reported", you must add references, in which articles it has been reported. Moreover, autoimmune disorders and aging are two different aspects of MSC dysfunction.
Page 5.
Line 200
Bailey, C et all, report that many viruses, including dengue virus, Ebola virus, influenza A virus, West Nile virus, but not SARS CoV2 change the expression of interferon-stimulated genes.
In the paper entitled "Human Mesenchymal Stromal Cells Are Resistant to SARS-CoV-2 Infection under Steady-State, Inflammatory Conditions and in the Presence of SARS-CoV-2-Infected Cells", Schäfer R et all, (PMID: 32950067) show the effect of SARS CoV2 on MSC. Please add this reference and rephrace the text.
In addition, human MSC do not express ACE2 and TMPRSS2 and are not permissive to SARS-CoV-2 infection, as it was reported by Avanzini MA et. al.
Which mechanism make the MSC resistant to viral infection, the expression of interferon-stimulated genes (ISG) or down-regulation of expression ACE2 and TMPRSS2 on the cells?
line 203
With the phrase "On the other hand" we usually describe the opposite effect, but in this sentence, you describe by which mechanisms the MSC promote their antimicrobial effect. Please rephrase this paragraph.
Line 220
Please add full dot after the ref 33 and start with new sentence to describe this experimental model.
Line 222
Celltype are two words. Please add space after cell.
Page 6
Line 244
in vivo should be set in italics.
Line 248
Please delete quotation marks before Recently.
Page 9
The section 6 does not reflect to the title "MSC-Based Therapy in COVID-19". It is better to add one general title about the COVID-19, because you describe all aspects of diseases, including the epidemiology, pathogenesis and therapy, not only MSC-Based Therapy in COVID-19.
This title may be used as subsection of this large section.
Author Response
Dr. Francisco Vizoso
Unidad de Investigación
Fundación Hospital de Jove
Avda. Eduardo Castro, 161
33290 Gijón, SPAIN
Fax: +34 985103564
E-mail: investigacion@hospitaldejove.com
July 17th, 2021
Dear Editor,
Please find enclosed the revised version of our manuscript (ijms-1312284) where we have responded to all the reviewer's comments. We thank you and Reviewers for the valuable criticisms and suggestions, which will undoubtedly translate into an improved version of the review.
Our specific answers to the reviewer’s comments are listed below.
Reviewer 2
We are grateful for the comments and very thorough analysis of the reviewer, who has contributed frankly to improve the manuscript. It is a luxury for us to have your observations and translate them into the modified version of the manuscript.
Page 2
Line 57
in vivo should be set in italics.
According to the reviewer indication, we changed de expression "in vivo" by *in vivo in the modified version of the manuscript. Introduction, page 2, line 59.
Line 69
Please correct word Resucitation as Resuscitation
According to reviewer suggestion, we changed the expression “Resucitation” by *Resuscitation in the modified version of the manuscript. Introduction, page 2, line 71.
Page 3
Line 127
Please add reference for paper by Friedenstein et al.
Following the suggestion from the reviewer, we included the reference for paper by Friedenstein et al., corresponding to the reference number 16. Section 3., Page 3, line 133.
Page 4
Line 152
If you start the paragraph with "it has been reported", you must add references, in which articles it has been reported. Moreover, autoimmune disorders and aging are two different aspects of MSC dysfunction.
According to the reviewer's observations, we incorporated the references that support MSC depletion and dysfunction in the different stated diseases, as can be seen in the modified version of the manuscript. References: Rheumatoid arthritis: 20; Systemic lupus erythematosus: 21; Diabetes Mellitus: 22; Psoriasis: 23. Subsection 3., page 4, line 159.
In addition, we have separated the concepts of autoimmune pathologies and aging, with the references provided given its pertinence, corresponding to each differentiated process, as can be seen in the modified version of the manuscript. References: Aging: 24, 25. Subsection 3., page 4, line 160.
Page 5
Line 200
Bailey, C et all, report that many viruses, including dengue virus, Ebola virus, influenza A virus, West Nile virus, but not SARS CoV2 change the expression of interferon-stimulated genes.
We greatly appreciate the reviewer's input on this point. As the reviewer mentioned, SARS-CoV-2 virus does not change the expression of interferon-stimulated genes. Therefore, SARS-CoV-2 has been excluded from the list of viruses for which this mechanism is valid, and we have added the sentence: “…makes them usually resistant to viral infections, such as dengue virus, Ebola virus, influenza A virus and West Nile virus, inter alia”, as can be seen in the edited version of the review. 4.1. Subsection, page 5, line 210.
In the paper entitled "Human Mesenchymal Stromal Cells Are Resistant to SARS-CoV-2 Infection under Steady-State, Inflammatory Conditions and in the Presence of SARS-CoV-2-Infected Cells", Schäfer R et all, (PMID: 32950067) show the effect of SARS CoV2 on MSC. Please add this reference and rephrace the text.
In conformity with the examiner's suggestion, we included the reference of Schäfer R et al. in the text (reference 36) and we rephrased the sentence regarding the effect of SARS-CoV-2 on MSC: “However, the infection capacity of SARS-CoV-2 may be due to IFN-independent pathways that could regulate the expression of currently known ACE2 or TMPRSS2 virus input receptors”, as can be seen in the modified version of the manuscript. 4.1. Subsection, page 5, line 210.
In addition, human MSC do not express ACE2 and TMPRSS2 and are not permissive to SARS-CoV-2 infection, as it was reported by Avanzini MA et. al.
Which mechanism make the MSC resistant to viral infection, the expression of interferon-stimulated genes (ISG) or down-regulation of expression ACE2 and TMPRSS2 on the cells?
In conformity with the reviewer's suggestion, we included the reference of Avanzini MA et. in the text (reference 37) and weclarified the mechanism of resistance of MSC to the SARS-CoV-2 virus adding the sentence “Continuing with the preceding, the very weak or non-existent expression of ACE2 and TMPRSS2 receptors on the surface of MSC, has been categorically demonstrated, making MSC theoretically resistant to SARS-Cov-2 infection. For that reason, alternative routes of entry for SARS-CoV-2 in MSC cannot be excluded, so more studies are needed to testify the evidence that human MSC are not lenient for SARS-CoV-2 infection” as can be seen in the modified version of the manuscript. 4.1. Subsection, page 5, line 213.
Line 203
With the phrase "On the other hand" we usually describe the opposite effect, but in this sentence, you describe by which mechanisms the MSC promote their antimicrobial effect. Please rephrase this paragraph.
According to the reviewer indication, we changed de expression "on the other hand" by *alternatively in the modified version of the manuscript. 4.1. Subsection, page 5, line 219.
Line 220
Please add full dot after the ref 33 and start with new sentence to describe this experimental model.
In accordance with the reviewer's request, we added full dot after reference 33 (now 42) and then started a new sentence describing the experimental model: “Analogously, some researchers have also pointed out that MSC have beneficial effects on experimental models for ALI, such as reducing pulmonary edema and maintaining alveolar-endothelial barrier homeostasis, actions mediated by TGF-β1, IDO, NO, IL-1RA, KGF and IL-10.”, as can be seen in the modified version of the manuscript. 4.2. Subsection, page 6, line 240.
Line 222
Celltype are two words. Please add space after cell.
In accordance with the reviewer's request, we added space after “cell”, now displaying the term *cell type in the modified version of the manuscript. 4.2. Subsection, page 6, line 244.
Page 6
Line 244
in vivo should be set in italics.
According to the reviewer indication, we changed de expression "in vivo" by *in vivo in the modified version of the manuscript. 4.4. Subsection, page 6, line 267.
Line 248
Please delete quotation marks before Recently.
In accordance with the reviewer's request, we deleted quotation marks before the term “Recently”. 4.4. Subsection, page 6, line 271.
Page 9
The section 6 does not reflect to the title "MSC-Based Therapy in COVID-19". It is better to add one general title about the COVID-19, because you describe all aspects of diseases, including the epidemiology, pathogenesis and therapy, not only MSC-Based Therapy in COVID-19.
This title may be used as subsection of this large section.
Based on the reviewer's comments, we added a general title to section 6 on COVID-19: “COVID-19: Facts and challenges” (Section 6., page 9, line 389) and several subsections regarding: epidemiology (“Epidemiology”, Subsection 6.1., page 9, line 390), pathogenesis (“Pathogenesis”, Subsection 6.2., page 9, line 404) and therapy (“MSC-Based Therapy in COVID-19", Subsection 6.3., page 11, line 512), as can be seen in the modified version of the manuscript.
Round 2
Reviewer 2 Report
All my previous remarks for this paper were corrected by authors.